

# Copula-based synthetic data generation for machine learning emulators in weather and climate: application to a simple radiation model

David Meyer[1,2], Thomas Nagler[3], Robin J. Hogan[4,1]

[1]Department of Meteorology, University of Reading, Reading, UK, [2]Department of Civil and Environmental Engineering, Imperial College London, London, UK, [3]Mathematical Institute, Leiden University, Leiden, The Netherlands, [4]European Centre for Medium-Range Weather Forecasts, Reading, UK

Correspondence to David Meyer (d.meyer@pgr.reading.ac.uk)

## Abstract

Can we improve machine learning (ML) emulators with synthetic data? The use of real data for training ML models is often the cause of major limitations. For example, real data may be (a) only representative of a subset of situations and domains, (b) expensive to source, (c) limited to specific individuals due to licensing restrictions. Although the use of synthetic data is becoming increasingly popular in computer vision, the training of ML emulators in weather and climate still relies on the use of real data datasets. Here we investigate whether the use of copula-based synthetically-augmented datasets improves the prediction of ML emulators for estimating the downwelling longwave radiation. Results show that bulk errors are cut by up to 75 % for the mean bias error (from 0.08 to -0.02 W m$^{-2}$) and by up to 62 % (from 1.17 to 0.44 W m$^{-2}$) for the mean absolute error, thus showing potential for improving the generalization of future ML emulators.



## 1. Introduction

The use of machine learning (ML) in weather and climate is becoming increasingly relevant (Huntingford et al., 2019; Reichstein et al., 2019). Two main strategies are currently identified for training ML models: one where input and output pairs are provided, and a second where inputs are provided, and outputs are generated using a physical model; here we define the former as observation-based training (OBT) and the latter as emulation-based training (EBT). Although OBTs are the most common training strategy currently used in ML, EBTs allow the creation of fast surrogate ML models (or emulators) 25 to replace complex physical parameterisation schemes (e.g. Chevallier et al., 1998; Krasnopolsky et al., 2002; Nowack et al., 2018).

In ML, the best way to make a model more generalizable is to train it on more data (Goodfellow et al., 2016). Although this is fairly easy to do for classification tasks (e.g. by translating or adding noise to an image), this may not be the case for most 30 regression tasks found in weather and climate. In this context, it is common to work with high dimensional and strongly dependent data (e.g. between physical quantities such as air temperature, humidity, and pressure across grid points), and although this dependence may be well approximated by physical laws (e.g. the ideal gas law for conditions found in the Earth's atmosphere), the generation of representative data across multiple dimensions is challenging.

To serve a similar purpose to that of real data, synthetically generated data need to preserve the statistical properties of real data in terms of the individual behaviour and (inter-)dependences. Several methods may be suitable for generating synthetic data generation such as copulas (e.g. Patki et al., 2016), variational autoencoders (e.g. Wan et al., 2017) and, more recently, generative adversarial networks (GANs; e.g. Xu and Veeramachaneni, 2018). Although the use of GANs for data generation is becoming increasingly popular amongst the core ML community, these require multiple models to be trained, leading to 40 difficulties and computational burden (Tagasovska et al., 2019). Variational approaches, on the other hand, make (strong) distributional assumptions, potentially detrimental to the generative model (Tagasovska et al., 2019). Compared to black-box deep learning models, the training of (vine) copulas is relatively easy and robust, while taking away a lot of guesswork in specifying hyperparameters and network architecture. Furthermore, copula models give a direct representation of the statistical distribution, which makes them easier to interpret and tweak after training. As such, the use of copula-based 45 models have been effective in generating synthetic data that are very close to the real data (Patki et al., 2016) in the context of privacy protection.

The goal of this paper is to determine whether training ML models with synthetically augmented datasets improves predictions. Here, we first summarize and formalize four main strategies identified to train ML models in a method that may





be generalizable beyond the scope of this paper (section 2) and implement it using a simple radiation physical, copula and

ML model (sections 2.2-2.6). We then evaluate results using separate error metrics for copula and ML models (section 3) and

report them (section 4) before concluding with a discussion and prospects for future research (section 5).

## 2.  Material and methods

### 2.1  Overview

The general method for *training* a ML model involves the use of paired *inputs* $X = \{x_1, \ldots, x_n\}$ and *outputs* $Y = \{y_1, \ldots, y_n\}$

to produce *weights* $w$ that correspond to the best function approximation for a specific model architecture and

configuration. For *inference*, the trained ML model uses the previously learned weights $w$ to predict new outputs $Y^*$ from

unseen inputs $X^*$. In the context of weather and climate, two main ML training strategies may be identified: observation-

based training (OBT; Figure 1 A and B) and emulation-based training (EBT; Figure 1 C and D). In the former, both $X$ and $Y$ are

used to train the ML model. In the latter, $Y$ is first generated with a physical model from $X$, and fed to the ML model for

training. Although OBT strategies are more common, EBT may be useful to create surrogate ML models (i.e. emulators) that

are faster, but only slightly less accurate, than their physical counterparts (e.g. Chevallier et al., 1998). In this paper we

introduce an additional step, that is, the generation of synthetic data (Figure 1 B and D), with the goal of improving the

prediction of ML models. We define a general methodology (Figure 1) for training ML models using OBT and EBT strategies,

and with (Figure 1 B and D) or without (Figure 1 A and D) data generation, as follows:

A.  **OBT**: Standard method for training ML models. Inputs $X$ and outputs $Y$ are used to train the ML model (Figure 1 A).

B.  **OBT with data generation**: Generation of synthetic samples for training ML models. A data generation model (here

copula) is fitted to both inputs $X$ and outputs $Y$ to generate synthetic inputs $X'$ and outputs $Y'$. $X'$ and $Y'$ are used to

train the ML model (Figure 1 B).

C.  **EBT**: Standard method for training ML emulators. Inputs $X$ are fed to the physical model to generate corresponding

outputs $Y$. $X$ and $Y$ used to train the ML model (Figure 1 C).

D.  **EBT with data generation**: Generation of synthetic samples for training ML emulators. A data generation model (here

copula) is fitted to inputs $X$ only to generate synthetic inputs $X'$. Inputs $X'$ are fed to the physical model to generate

corresponding outputs $Y'$. $X'$ and $Y'$ are used to train the ML model (Figure 1 D).



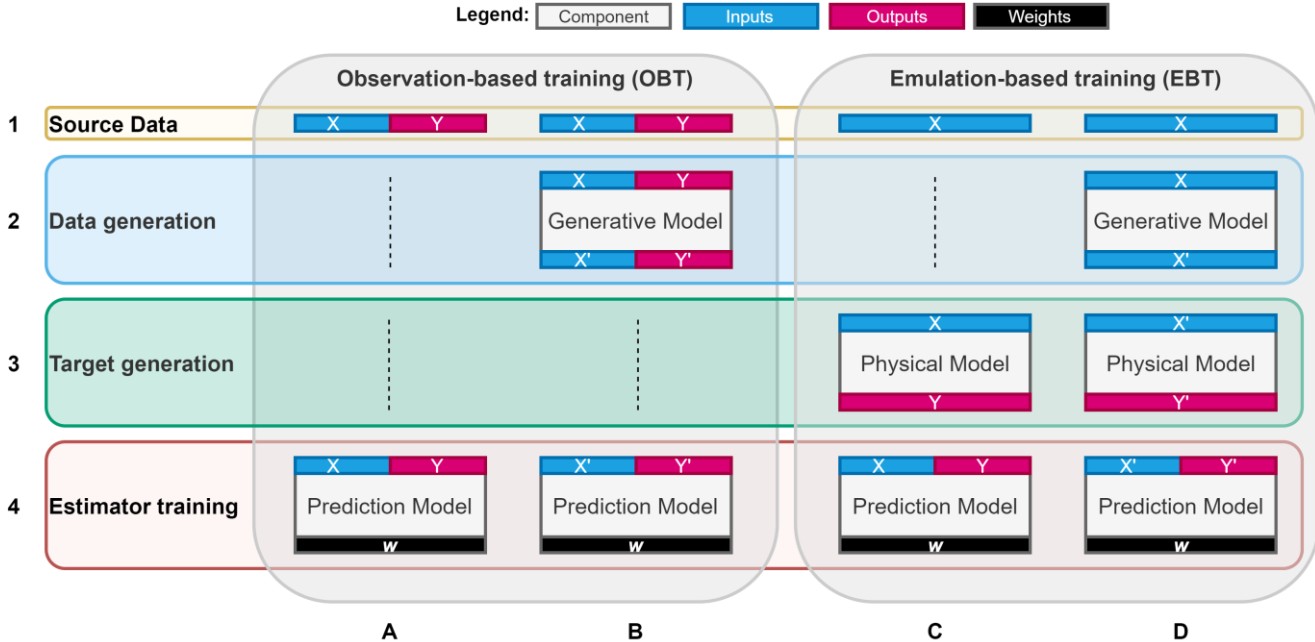

**Figure 1. Main strategies identified for training machine learning (ML) models in weather and climate: (A) traditional method for training using input and output pairs and (C) if a physical model is available (model emulation) the corresponding output targets are generated by a physical model. B and D are the same as A and C respectively with the addition of data generation (this paper).**

To evaluate whether ML models trained with both real and synthetic data (i.e. B and D) have a lower prediction error than those trained with only the real data (i.e. A or C), here we focus on the prediction of vertical profiles of longwave radiation from those of dry-bulb air temperature, atmospheric pressure, and cloud optical depth. This task is chosen at it allows us to: (i) evaluate copula-based models for generating correlated multidimensional data (e.g. with dependence across several quantities and grid points), some of which (e.g. cloud optical depth) are highly non-Gaussian; (ii) develop a simple and fast toy physical model that may be representative of other physical parameterizations such as radiation, (urban) land surface, cloud, or convection schemes; and (iii) develop a fast and simple ML model used to compute representative statistics. We then define case A (or C) as the *baseline* and generate six different subcases from case B and D, each using (i) three levels of data *augmentation factors* (i.e. either 1x, 5x or 10x the number of profiles in the real dataset), (ii) generated from three different copula classes.

In the following sections we give background information and specific implementation details about the general method used for setting up the source data (section 2.2), data generation (section 2.3), target generation (section 2.4), and estimation training (section 2.5) as shown in Figure 1.



## 2.2 Source Data

Depending on the strategy used, source data may (i) be used as input to the prediction, generative, or physical model, (ii)
100    contain input and output pairs or inputs only, (iii) consist of real or synthetically generated data (Figure 1). Furthermore,
depending on whether source data are used for training or for inference, different subsets may be used at different times.

Here, we define a *source dataset* derived from the EUMETSAT Numerical Weather Prediction Satellite Application Facility
(NWP-SAF) dataset (Eresmaa and McNally, 2014). The NWP-SAF is a dataset of common meteorological variables used to
105    evaluate the performance of radiation models (e.g. Hogan and Matricardi, 2020). It contains a representative collection of
25 000 vertical profiles of the atmosphere from global operational short-range ECMWF (European Centre for Medium-Range
Weather Forecasts) forecasts for 137 vertical levels, correlated in more than one dimension (between quantities and spatially
across levels), and extending from top of the atmosphere (TOA; 0.01 hPa; level 1;) to the surface (bottom of the atmosphere;
BOA; level 137). Here, to compare OBT and EBT strategies, we create inputs $X$ and outputs $Y$ partitions (Table 1) as follows:
$X$ contains vertical profiles of dry-bulb air temperature ($T$ in K; Figure 2a), atmospheric pressure ($p$ in hPa; Figure 2b), and
derived layer cloud optical depth ($\tau_c$; Figure 2c) from other variables in the NWP-SAF dataset to simplify the creation of
models described in this paper (section 2.4); $Y$ contains vertical profiles of downwelling longwave radiation ($L^{\downarrow}$ in W m$^{-2}$;
Figure 2d) computed from the physical model (section 2.4). We then use $X$ and $Y$ in OBT strategies (Figure 1 A and B) and
only $X$ in EBT (Figure 1 C and D). Prior to be used, the source dataset is shuffled at random and split into three batches of 10
000 profiles (40 %) for *training* ($X_{\text{train}}$, $Y_{\text{train}}$), 5 000 (20 %) for *validation* ($X_{\text{validation}}$, $Y_{\text{validation}}$), and 10 000 (40 %) for *testing*
($X_{\text{testing}}$, $Y_{\text{testing}}$) and referred to as such throughout the paper. Furthermore, as both copula and ML models work on two-
dimensional data, datasets are converted to a matrix with samples as rows and flatten profiles per quantities as columns. To
compute plots and statistics, the data are reconstructed to their original shape.


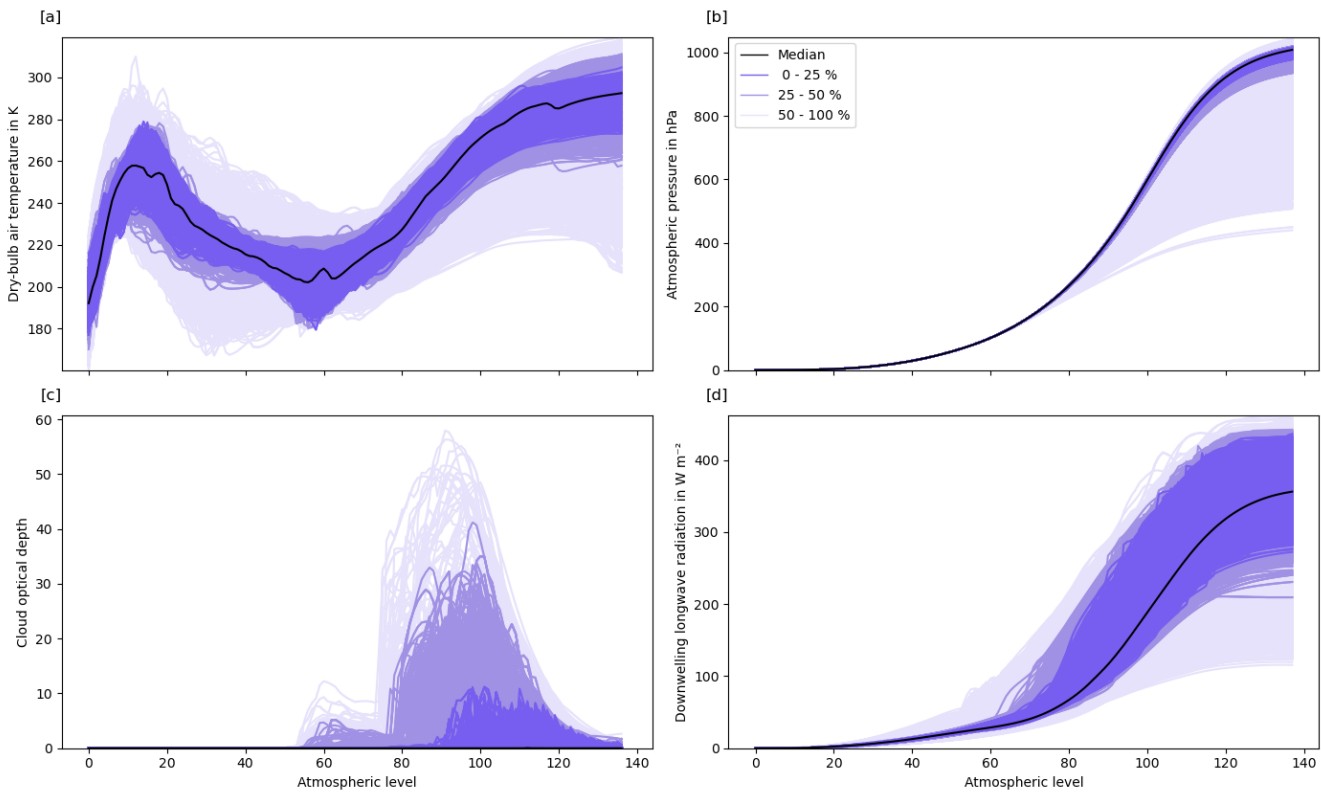

**Figure 2.** Profiles of (a) dry-bulb air temperature, (b) atmospheric pressure, (c) cloud optical depth, (d) downwelling longwave radiation
from the NWP-SAF dataset (25 000 profiles; Eresmaa and McNally, 2014). Profiles are ordered using band depth statistics (López-Pintado
and Romo, 2009) and shown for their most central (median) profile and grouped for the central 0–25 %, 25 − 50 % and 50 − 100 %.

**Table 1.** Profiles of input and output quantities used in this study. Input quantities are dry-bulb air temperature $T$, atmospheric
temperature $p$ and cloud optical depth $\tau_c$. $T$ and $p$ are taken directly from the NWP-SAF dataset (Eresmaa and McNally, 2014). $\tau_c$ is
derived from other quantities as described in section 2.4. The output quantity is downwelling longwave radiation $L^{\downarrow}$ and is computed
using the physical model described in section 2.4. The number of atmospheric model levels is 137 for full levels (FL) and 138 for half
levels (HL).

| Symbol | Name | Unit | Dimension |
|---|---|---|---|
| (a) Inputs | | | |
| $T$ | Dry-bulb air temperature | K | FL |
| $p$ | Atmospheric pressure | Pa | FL |
| $\tau_c$ | Cloud optical depth | 1 | FL |
| (b) Output | | | |
| $L^{\downarrow}$ | Downwelling longwave radiation | W m$^{-2}$ | HL |





## 2.3 Data generation

Data generation is used to generate additional samples (here the atmospheric profiles) to be fed to the physical (section 2.4) and ML (section 2.5) model. Optimally, these synthetically generated data should resemble the observed data as closely as possible with regard to (i) the individual behaviour of variables (e.g. the dry-bulb air temperature at a specific level), and (ii) the dependence across variables and dimensions (e.g. the dry-bulb air temperature across two levels). Copulas are statistical models that allow to disentangle these two aims (Trivedi and Zimmer, 2006; Joe, 2014) and to generate new samples that

are statistically similar to the original data in terms of their individual behaviour and dependence.

### 2.3.1 Background on copula models

Suppose we want to generate synthetic data from a probabilistic model for $d$ variables $Z_1, \dots, Z_d$. To achieve the first aim, we need to find appropriate *marginal distributions* $F, \dots, F_d$. A simple approach is to approximate them by the corresponding

empirical distribution functions. To achieve the second aim, however, we need to build a model for the *joint distribution function* $F(z_1, \dots, z_d)$. The key result, Sklar's theorem (Sklar, 1959), states that any joint distribution function can be written as

$$F(z_1, \dots, z_d) = C(F_1(z_1), \dots, F_d(z_d)).$$

The function $C$ is called copula and encodes the dependence between variables.


Copulas are distribution functions themselves. More precisely, if all variables a continuous, $C$ is the joint distribution of the variables $U_1 = F_1(Z_1), \dots, U_d = F_d(Z_d)$. This fact facilitates estimation and simulation from the model. To estimate the copula function $C$, we (i) estimate marginal distributions $\widehat{F}_1, \dots, \widehat{F}_d$, (ii) construct *pseudo-observations* $\widehat{U}_1 = \widehat{F}_1(Z_1), \dots, \widehat{U}_d = \widehat{F}_d(Z_d)$, and (iii) estimate $C$ from the pseudo-observations. Then, given estimated models $\widehat{C}, \widehat{F}_1, \dots, \widehat{F}_d$

for the copula and marginal distributions, we can generate synthetic data as follows:

1. Simulate random variables $U_1, \dots, U_d$ from the estimated copula $\widehat{C}$.

2. Define $Z_1 = \widehat{F}_1^{-1}(X_1), \dots, Z_d = \widehat{F}_d^{-1}(X_d)$.





### 2.3.2 Parametric copula families

In practice, it is common to only consider sub-families of copulas that are conveniently parametrized. There is a variety of such parametric copula families. Such families can be derived from existing models for multivariate distributions by inverting the equation of Sklar's theorem:

$$C(u_1, \dots, u_d) = F(F_1^{-1}(u_1), \dots, F_d^{-1}(u_d)).$$

For example, we can take $F$ as the joint distribution function of a multivariate Gaussian and $F_1, \dots, F_d$ as the corresponding
marginal distributions. Then the display above yields a model for the copula called *Gaussian copula,* which is parametrized by a correlation matrix. The Gaussian copula model subsumes all possible dependence structure in a multivariate Gaussian distribution. The benefit comes from the fact that we can combine a given copula with any type of marginal distributions, not just the ones the copula was derived from. That way, we can build flexible models with arbitrary marginal distributions and Gaussian-like dependence. The same principle applies to other multivariate distributions and many copula models have
been derived, most prominently the Student t copula and Archimedean families. A comprehensive list can be found in Joe (2014).

### 2.3.3 Vine copula models

When there are more than two variables ($d > 2$) the types of dependence structures these models can generate is rather
limited. Gaussian and Student copulas only allow for symmetric dependencies between variables. Quite often, dependence is asymmetric, however. For example, dependence between $Z_1$ and $Z_2$ may be stronger when both variables take large values. Many Archimedean families allow for such asymmetries but require all pairs of variables to have the same type and strength of dependence.

Vine copula models (Aas et al., 2009; Czado, 2019) are a popular solution to this issue. The idea is to build a large dependence model from only two-dimensional building blocks. We can explain this with a simple example with just three variables $Z_1, Z_2, Z_3$. We can model the dependence between $Z_1$ and $Z_2$ by a two-dimensional copula $C_{1,2}$ and the dependence between $Z_2$ and $Z_3$ by another, possibly different, copula $C_{2,3}$. These two copulas already contain some information about the dependence between $Z_1$ and $Z_3$, the part of the dependence that is induced by $Z_2$. The missing piece is the dependence
between $Z_1$ and $Z_3$ after the effect of $Z_2$ has been removed. Mathematically, this is the conditional dependence between $Z_1$ and $Z_3$ given $Z_2$ and can be modeled by yet another two-dimensional copula $C_{1,3|2}$. The principle is easily extended to an arbitrary number of variables $Z_1, \dots, Z_d$. Algorithms for simulation and selecting the right conditioning order and parametric families for each (conditional) pair are given in Dißman et al. (2013).





Because all two-dimensional copulas can be specified independently, such models are extremely flexible and allow for highly

heterogenous dependence structures. Using parametric models for pair-wise dependencies remain a limiting factor,

however. If necessary, it is also possible to use nonparametric models for the two-dimensional building blocks. Here, the

joint distribution of pseudo-observations $\widehat{U}_1$, $\widehat{U}_2$ is estimated by a suitable kernel density estimator (see Nagler et al., 2017).

### 2.3.4 Implementation


Here we use Synthia (Meyer and Nagler, 2020), to fit three different copula types: Gaussian, Vine-parametric, Vine-

nonparametric. Each copula model is fitted to the training set $X_{\text{train}}$ in OBT, and to both, $X_{\text{train}}$ and $Y_{\text{train}}$ sets, in EBT. To

evaluate the impact of copula-augmented datasets on the ML inference, we generate synthetic profiles with augmentation

factors of 1x, 5x, and 10x the number of profiles included in the source training dataset (i.e. 10 000 profiles). These are then

used to create *augmented* versions of training datasets, here defined as $X'_{\text{train}}$ and $Y'_{\text{train}}$, each containing the source plus

the synthetically generated profiles (i.e. with 20 000, 60 000, or 110 000 profiles). As the generation of new profiles with

copula models is random, the generation is repeated 10 times for each case to allow for meaningful statistics to be computed.

### 2.4  Target generation

Target generation (Figure 1 C-D) is used in EBTs to generate outputs from corresponding inputs. Here, however, to compare

results from the two different strategies described in this paper (i.e. OBT vs EBT), we also use target generation to compute

outputs for the source dataset $Y$ in OBT strategies. In all cases, outputs $Y$ are computed using a simple toy model based on

Schwarzschild's equation (e.g. Petty, 2006) to estimate the downwelling longwave radiation under the assumption that

atmospheric absorption does not vary with wavelength, as:


$$\frac{dF}{dz} = a(z)[B(z) - F] \qquad (1)$$

where $z$ is the geometric height, $B$ is the Planck function at the temperature at level $z$ (i.e. $B = \sigma_{\text{SB}}T^4$, where $\sigma_{\text{SB}}$ is the

Stefan-Boltzmann constant; giving the flux in W m$^{-2}$ emitted from a horizontal black body surface), and $a$ is the rate at which

radiation is intercepted/emitted.  A common approximation is to treat longwave radiation travelling at all angles as if it were

all travelling with a zenith angle of 53 degrees (Elsasser, 1942): in this case $a = D\beta_e$ where $\beta_e$ is the extinction coefficient of

the medium, and $D = 1.66 = 1/\cos(53)$ is the diffusivity factor, which accounts for the fact that the effective path length





of radiation passing through a layer of thickness $\Delta z$ is on average $1.66\Delta z$ due to the multiple different angles of propagation. In the context of ML, $a(z)$ and $B(z)$ are known and $F(z)$ is to be predicted. Here we use the difference in two atmospheric

pressures expressed in sigma coordinates ($\Delta\sigma$, where $\sigma$ is the pressure $p$ at a particular height divided by the surface pressure $p_0$) instead of $z$. The cloud layer optical depth $\tau = \beta_e \Delta z$ is calculated from the total column gas optical depth $\tau_g$ and layer cloud optical depth $\tau_c$ as $\tau = \tau_c + \tau_g \Delta\sigma_i$ as $\Delta\sigma$ is the fraction of mass of the full atmospheric column in layer $i$. Then, as the downwelling flux at the top of the atmosphere is 0, the equation is discretized as follows assuming $B$ and $a$ are constant within a layer:


$$F_{i-1/2} = F_{i+1/2}\,(1 - \epsilon_i) + B_i\epsilon_i, \tag{2}$$

where $B_i$ is the Planck function of layer $i$, $\epsilon_i = 1 - e^{-a_i\Delta z} = 1 - e^{D\tau}$ is the emissivity of layer $i$, $F_{i+1/2}$ is the downwelling flux at the top of layer $i$, and $F_{i-1/2}$ is the downwelling flux at the bottom of layer $i$. We compute $L^{\downarrow}$ in W m$^{-2}$ from $T$ in K, $p$

in Pa, and $\tau_c$ using the source $X$ or augmented $X'$ data depending on the strategy (i.e. OBT or EBT). To reduce, and thus simplify, the number of quantities used in the physical and ML models (section 2.5), $\tau_c$ is pre-computed and used instead of vertical profiles of liquid and ice mixing ratios ($q_l$ and $q_l$ in 1) and effective radius ($r_l$ and $r_l$ in m) as $\frac{3}{2}\frac{\Delta p}{g}\left(\frac{q_l}{\rho_l r_l} + \frac{q_i}{\rho_i r_i}\right)$, where $\rho_l$ is the density of liquid water (1 000 kg m$^{-3}$), $\rho_i$ is the density of ice (917 kg m$^{-3}$), $g$ is the standard gravitational acceleration (9.81 m s$^{-2}$). For $\tau_g$ we use a constant value of 1.7 determined by minimizing the absolute error between profiles computed

with this simple model and the comprehensive atmospheric radiation scheme ecRad (Hogan and Bozzo, 2018).

## 2.5 Estimator training

As the goal of this paper is to determine whether the use of synthetic data improves the prediction of ML models, here we implement a simple feedforward neural network (FNN). FNNs are one of the simplest and most common neural networks used in ML (Goodfellow et al., 2016) and have been previously used for similar weather and climate applications (e.g.

Chevallier et al., 1998; Krasnopolsky et al., 2002). FNNs are composed of artificial neurons (conceptually derived from biological neurons) connected with each other where information moves forward from the input nodes, through hidden nodes. The multilayer perceptron (MLP) is a type of FNN composed of at least three layers of nodes: an input layer, a hidden layer, and an output layer with all but the input nodes using a nonlinear activation function.


Here we implement a simple an MLP consisting of 3 hidden layers with 512 neurons each. This is implemented in TensorFlow (Abadi et al., 2015), and configured with elu activation function, Adam optimizer, Huber loss, 1 000 epochs limit, and early





stopping with patience of 25 epochs. The MLP is trained with profiles of dry-bulb air temperature ($T$ in K; Figure 2a), atmospheric pressure ($p$ in hPa; Figure 2b), and layer cloud optical depth ($\tau_c$; Figure 2c) as inputs and profiles of longwave

downwelling longwave radiation ($L^{\downarrow}$ in W m$^{-2}$; Figure 2d) as outputs. Inputs are normalized and both inputs and outputs are flattened into feature vectors. The baseline case (Figure 1 A or C) use 10 000 input profiles without data augmentation (i.e. using $X_{\text{train}}$ and $Y_{\text{train}}$) for training and copula-based cases (Figure 1 B and D) use either 20 000, 60 000, or 110 000 profiles (i.e. using $X'_{\text{train}}$ and $Y'_{\text{train}}$). The validation dataset $Y_{\text{validation}}$ of 5 000 profiles is used as input for the early stopping mechanism while the test dataset $Y_{\text{test}}$ of 10 000 profiles is used to compute the error statistics using evaluation metrics

described in section 3.2. Because of the stochastic nature of the MLP used, training and inference is repeated 10 times for each case to allow for meaningful statistics to be computed. Given that the generation of random profiles in the case of augmented datasets ($X'_{\text{train}}$ and $Y'_{\text{train}}$) is also repeated 10 times (see section 2.3.4) all cases using data generation comprise of 100 iterations in total (i.e. for each data generation run, we run the ML fitting 10 times).

**3.    Evaluation metrics**

We conduct a twofold evaluation: first we assess the quality of synthetic data produced by different copula classes (section 3.1), then we assess the prediction error of ML model (section 3.2) trained using different augmentation factors. Although the former may be of interest to determine how well copula models may be used to generate profiles of different atmospheric quantities and to evaluate whether dependencies between variables have been captured, the latter is the main

focus here, used to evaluate whether ML models trained with augmented datasets of real and synthetic data have a lower prediction error than those trained with only the real data.

**3.1   Copula**

The quality of synthetic data is assessed in terms of summary statistics (e.g. Seitola et al., 2014) between the training $X_{\text{train}}$

and $Y_{\text{train}}$ and copula-simulated $X'_{\text{train}}$ and $Y'_{\text{train}}$ dataset. As the quality of the fitting may be different between the two strategies used, we compute separate statistics for OBT ($X_{\text{train}}$ and $Y_{\text{train}}$ vs $X'_{\text{train}}$ and $Y'_{\text{train}}$) and EBT ($X_{\text{train}}$ vs $X'_{\text{train}}$), i.e. the former having been fitted to both inputs and output pairs and the latter to only the inputs. For each copula type and training strategy, we compute a vector of summary statistics $S_i = f(\mathbf{P}_i)$ where $f$ is the statistic function and $\mathbf{P}_i = \mathbf{D}\boldsymbol{w}_i$, with $\mathbf{D}$ a matrix of flattened source or simulated data and $\boldsymbol{w}$ a vector of random numbers from the $i$th iteration. Summary

statistics are then computed for mean, variance, and quantiles, iterating 100 times to allow for meaningful statistics to be





computed. As we consider random linear combinations of variables in source and copula-generated data, we expect these summaries to coincide only if both marginal distributions and dependence between variables are captured.

### 3.2 Machine learning

The prediction error of the ML model is investigated by comparing outputs computed by the physical model with those computed at inference by the ML model fed with test dataset $X_{\text{test}}$ described in section 2.2. Here we use two common bulk error metrics to summarize errors across multiple profiles and atmospheric levels: mean bias error (MBE) and mean-absolute error (MAE). These are computed from a vector of random variables representing the differences, or error, $\boldsymbol{d} = (d_1, \dots, d_i)$ between the physically predicted $Y_{\text{test}}$ and ML predicted $Y'_{\text{test}}$ (i.e. $\boldsymbol{d} = Y_{\text{test}} - Y'_{\text{test}}$). Bulk error statistics are computed for the vector of outputs $1, \dots, N$ for the MBE and MAE (Table 2).


Table 2. Bulk error statistical metrics used in the machine learning evaluation. Mean bias error (MBE), mean-absolute error (MAE).

| MBE | MAE |
|---|---|
| $\dfrac{1}{N}\sum_{i=1}^{N} d_i$ | $\dfrac{1}{N}\sum_{i=1}^{N} |d_i|$ |

## 4 Results

### 4.1 Copula

We first check whether copula models can generate data that are statistically alike those in the source dataset. To this end, we compare summary statistics of random projections of generated and source data as described in section 3.1. Figure 3 shows scatterplots of summary statistics for their (a) mean, (b) variance, (c) standard deviation, and (d) 10 %, (e) 50 % and (f) 90 % quantiles. Summaries of the source data are on the x-axis while summaries of copula-generated data are on the y-axis. Each point corresponds to a random projection (100 points in total). For a perfect copula model, we expect all the 295 simulated points to fall on the main diagonal where $x = y$. Figure 3 shows that for all the copula models and configurations (with or without outputs) studied, the synthetically-generated data are similar to the real data, with larger errors in variance and standard deviation.

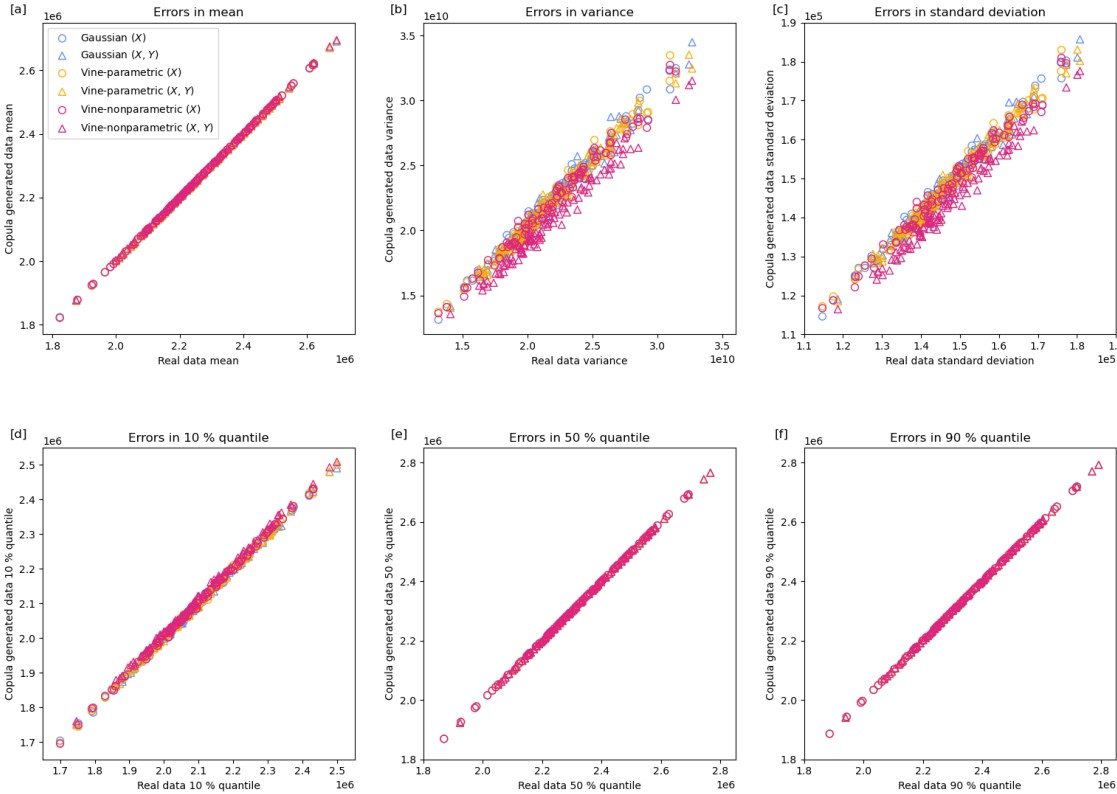

**Figure 3.** Summary statistics $S_i$ from 100 copula iterations for (a) mean, (b) variance, (c) standard deviation, and (d) 10 %, (e) 50 %, and (f) 90 % quantiles. Each point corresponds to a single iteration. Units are arbitrary. The x axis represents the projection of the true data $X_\mathrm{train}$ while the y axis that of the copula generated data $X'_\mathrm{train}$. Results reported for Gaussian, Vine-parametric, Vine-nonparametric copulas fitting to inputs or input and output pairs (i.e. $X_\mathrm{train}$ vs $X'_\mathrm{train}$ or $X_\mathrm{train}$ and $Y_\mathrm{train}$ vs $X'_\mathrm{train}$ and $Y'_\mathrm{train}$) – see legend.

Qualitatively, we can also evaluate copula-generated profiles in terms of their overall shape and smoothness across multiple levels, and range and density at each level. To this end we plot a side-by-side comparison of source (Figure 4left panel) and Gaussian-copula generated (Figure 4right panel) profiles showing the median profile and a random selection of 90 profiles grouped in batches of 3 (i.e. each having 30 profiles) for the central 0-25 % and outer 25-50 %, 50-100 % quantiles, calculated with band depth statistics (López-Pintado and Romo, 2009). Simulated profiles of dry-bulb air temperature (Figure 4b) appear less smooth than the real (Figure 4a) across levels; however, their density and range are simulated well at each level. Simulated profiles of atmospheric pressure (Figure 4d) are simulated well; they are smooth across all levels with a similar range and density than the real (Figure 4c). The highly non-Gaussian and spikey profiles of cloud optical depth (Figure 4e) make a qualitative comparison difficult, but the simulated profiles (Figure 4f) have a similar range and density, with high density for low values and most of the range between levels 80 and 120. Finally, copula-simulated profiles of downwelling longwave radiation (Figure 4h; only computed for OBT strategies) are noisier that the real (Figure 4g) but with a similar range and density.





**Figure 4.** Profiles of (left) real and (right) Gaussian copula-generated data of (a-b) dry-bulb air temperature, (c-d) atmospheric pressure, (e-f) cloud optical depth, (g-h) downwelling longwave radiation. Median profile shown in black and random selection of 90 profiles grouped in batches of 3 (i.e. each having 30 profiles) for the central 0-25 % and outer 25-50 %, 50-100 % calculated with band depth statistics (López-Pintado and Romo, 2009).

315





## 4.2 Machine learning

We report results for OBT and EBT strategies with or without data generation. Errors statistics are computed with metrics defined in section 3.2 against the test dataset of 10 000 profiles defined in section 2.2. Boxplots of bulk MBE and MAE are shown in Figure 5 for OBT (left) and for EBT (right). Summary bulk MBE and MAE for ML models with lowest MAE using an augmentation factor of 10x are reported in Table 3. A qualitative side-by-side comparison of MLP-generated profiles using Gaussian copula-generated profiles with augmentation factor of 10x and the corresponding baseline are shown in Figure 6.

MBEs in OBT (Figure 5a) are higher than the baseline across all copula models and augmentation factors, with median MBE and spread generally increasing with larger values of augmentation factors. Conversely, MBEs in EBT (Figure 5b) are generally lower than the baseline across all copula types and augmentation factors, with median MBE and spread decreasing with larger values of augmentation factors. MAEs in OBT (Figure 5c) do not improve from the baseline when additional synthetic data or different copula types are used. Overall, the Gaussian copula model performs better than the Vine-parametric or Vine-nonparametric models. This median MAE with 1x augmentation factor is approximately 2 W m$^{-2}$ for Gaussian, 2.4 W m$^{-2}$ for Vine-parametric and 2.6 W m$^{-2}$ for Vine-nonparametric, increasing with larger augmentation factors. Conversely to OBT, MAEs in EBT show a net improvement from the baseline across all copula models and augmentation factors (Figure 5d). When using an augmentation factor of 1x, the median MAE is reduced to approximately 1.1 W m$^{-2}$ using copula models from a baseline of approximately 1.4 W m$^{-2}$ and further reduced with increasing augmentation factors. In the best case, corresponding to an augmentation factor of 10x (i.e. with an additional 100 000 synthetic profiles added to the training source dataset), the copula and ML model combination producing the lowest values of MAE (Table 3) shows that both MBE and MAE are reduced from the baseline case. The MBE is reduced from a baseline of 0.08 W m$^{-2}$ to -0.02 and -0.05 W m$^{-2}$ for Gaussian and Vine-nonparametric respectively but increased to 0.10 W m$^{-2}$ for Vine-parametric. MAEs are reduced from a baseline of 1.17 W m$^{-2}$ to 0.45, 0.56 and 0.44 W m$^{-2}$ for Gaussian, Vine-parametric, Vine-nonparametric copula type respectively.

The ML training configuration to achieve the lowest overall MBE and MAE combination during inference correspond to a Gaussian copula and augmentation factor of 10x (Table 3). Differences (or errors) between the physically predicted $Y_{\text{test}}$ and ML predicted $Y'_{\text{test}}$ are shown for the baseline (Figure 6a) and Gaussian copula (Figure 6b). These are shown grouped by their central 0-25 % and outer 25-50 %, 50-100 %. Qualitatively most ML generated profiles show improvements from to the baseline. For the most central 25 % profiles are within ±20 W m$^{-2}$ for the Gaussian copula case, and about ±40 W m$^{-2}$ for the baseline case. Near surface errors (levels 130-BOA) are reduced to approximately ±5 W m$^{-2}$ from approximately ±10 W m$^{-2}$.



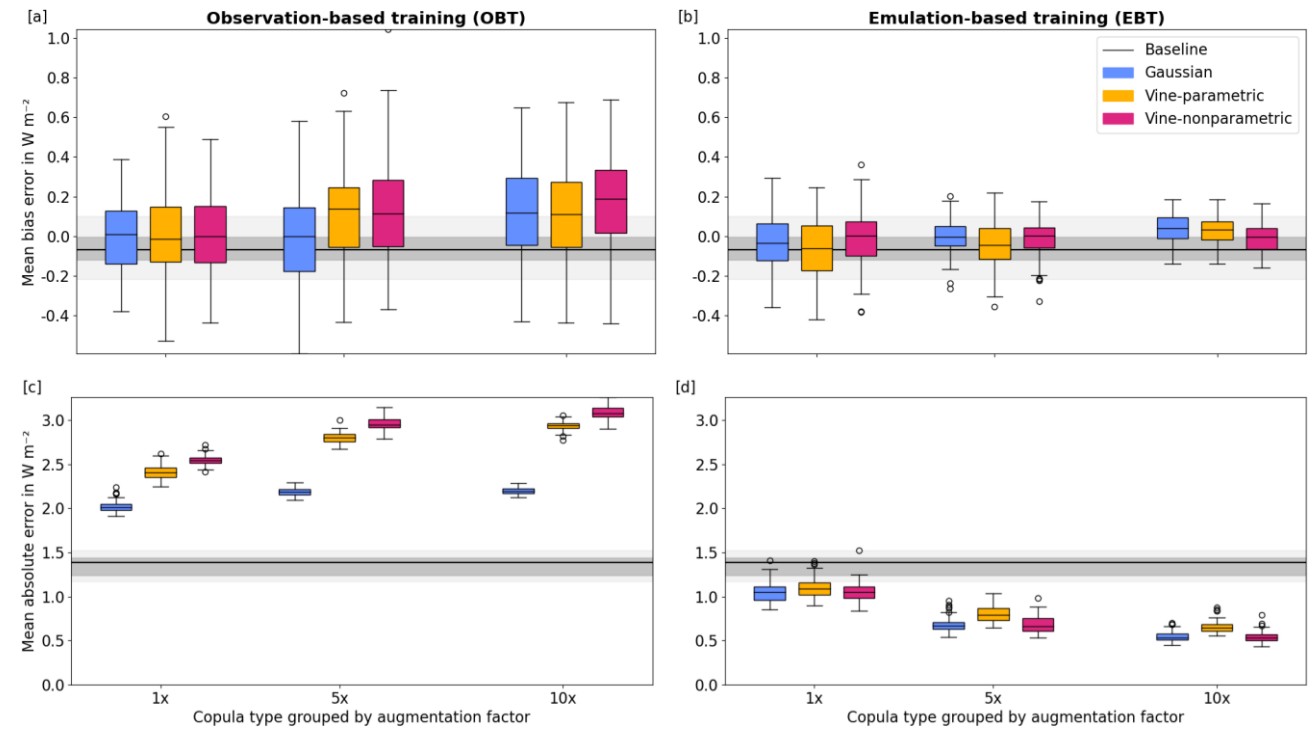

**Figure 5.** Training errors for (left) observation-based training and (right) emulation-based training grouped by different copula types (Gaussian: blue, Vine-parametric: yellow, Vine-nonparametric: red) and augmentation factors (1x, 5x, 10x) for the mean bias error (MBE; a-b) and mean absolute error (MAE; c-d). The median for the baseline case is shown in black and the range shaded in grey.

**Table 3.** Emulation-based training bulk mean bias error (MBE) and mean absolute error (MAE) for baseline ML model, and copula and ML model combination producing the lowest values of MAE. Baseline case trained using 10 000 real profiles and copula cases training using augmented dataset containing 110 000 profiles (10 000 real and 100 000 synthetic), i.e. with an augmentation factor of 10x.

| Case name | MBE in W m$^{-2}$ | MAE in W m$^{-2}$ |
|---|---|---|
| Baseline | 0.08 | 1.17 |
| Gaussian | **-0.02** | 0.45 |
| Vine-parametric | 0.10 | 0.56 |
| Vine-nonparametric | -0.05 | **0.44** |


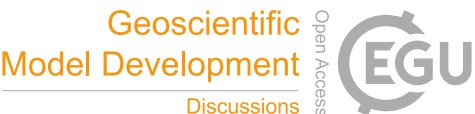

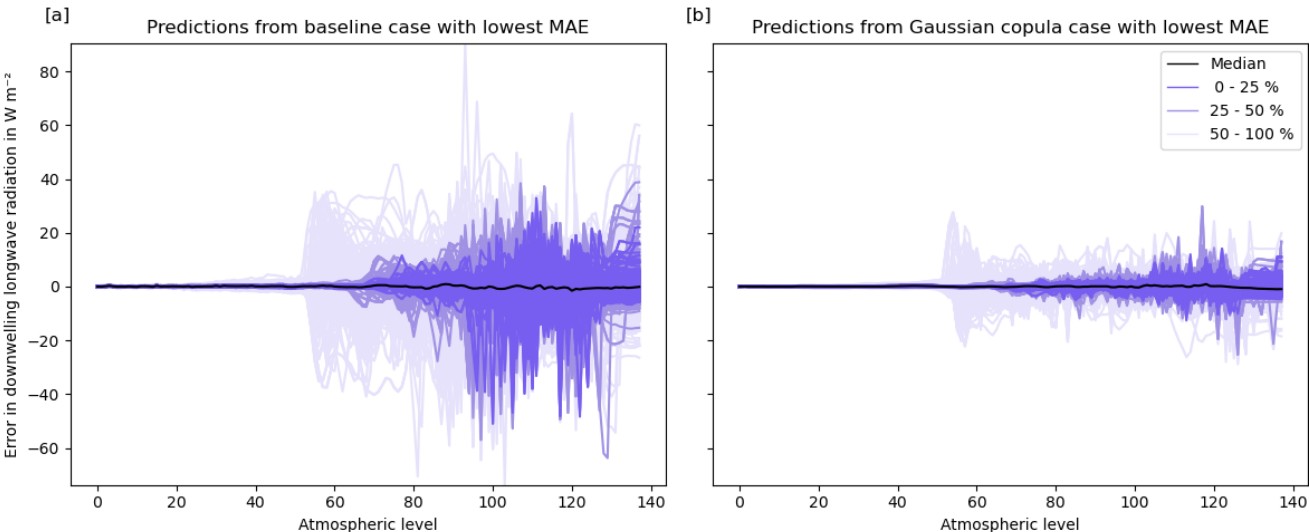

**Figure 6. ML errors in predicting downwelling longwave radiation for (a) baseline and (b) emulation-based training strategy using 110 000 profiles (10x augmentation factor; Gaussian copula). The median (most central) profile is shown in black and the most central 25 %, and outer 25 – 50 % and 50 – 100 % profiles are computed using band depth statistics and shown in shades of blue.**

## 5  Discussion and conclusion

Results from the machine learning evaluation show that bulk errors are cut by up to 75 % for the mean bias error (from 0.08 to -0.02 W m$^{-2}$;Table 3) and by up to 62 % (from 1.17 to 0.44 W m$^{-2}$;Table 3) for the mean absolute error in emulation-based training (EBT). This is not the case in observation-based training (OBT) where the use of synthetic data negatively affect the error (Figure 5). This finding is not surprising as model fits are merely an approximation of the real data and it is therefore unlikely to see improvements in predictions from OBT strategies from this or other type data generation methods (for type or model and configuration used). A qualitative comparison of synthetically generated profiles (Figure 4) shows that, although the main structure is captured, synthetic profiles tend to be less smooth and noisier than the real ones. This, together with the added complexity of having to capture the dependence between input and output pairs, may lead copula model to generate training samples that are too unrepresentative of the test data in the case of OBT strategies. On the other hand, when a physical model is available and an EBT strategy is used, (copula-based) data generation has the potential to improve error statistics by enriching the training dataset. In such cases, the dependence between inputs and outputs does not need to be captured as it is already modelled by the physical model. Instead, the data generation model needs to generate approximate inputs that are representative and valid for the physical model in use. In the simplest case, this may be, as simple as respecting the inverse relationship of pressure and temperature of ideal gasses or the positivity of absolute temperature.





Previous studies (e.g. Patki et al., 2016) have shown how copula-based models may be used to overcome data licensing restriction. Here we show how copula-based models may be used to improve the prediction of ML models in EBT strategies. This is done by generating augmented datasets containing statistically similar profiles in terms of their individual behavior and dependence across variables (e.g. dry-bulb air temperature at a specific level and across two levels). Although the focus of this paper is to evaluate copula-based data generation models that improve predictions, we speculate that the same or similar methods of data generation have the potential to be used in several other ML-related applications such as to: (i) test ML model architectures (e.g. instead of cross validation, one may generate synthetic datasets of different sizes to test the effect of the sample size on different ML architectures); (ii) generate data for un-encountered conditions (e.g. for climate change scenarios, by extending the range of the data, or relax marginal distributions); (iii) data compression (e.g. by storing reduced parameterized versions of the data if the number of samples is much larger than the number of features).

Although so far, we have only highlighted benefits of copula-based model, several limiting factors should be considered based on the specific problem and application. A key factor for very high-dimensional data is that both Gaussian and Vine copula models scale quadratically in the number of features – in terms of both memory and computational complexity. This can be alleviated by imposing structural constraints on the model, for example using structured covariance matrix or truncating the vine after a fixed number of trees. However, this limits their flexibility and adds some arbitrariness to the modelling process. A second drawback compared to GANs is that the model architecture cannot be tailored to a specific problem, like images. For such cases, a preliminary data compression step as in Tagasovska et al. (2019) may be necessary.

As highlighted here, data generation in EBT strategies may be of particular interest to scientists and practitioners looking to achieve a better generalization of their ML models (i.e. synthetic data may act as a regularizer to reduce overfitting; Shorten and Khoshgoftaar, 2019) and although a comprehensive analysis of prediction errors using different ML model architectures is out of scope, our work is a first step towards further research in this area. Moreover, although we did not explore the generation of data for un-encountered conditions (e.g. by extending the range of air temperature profiles while keeping a meaningful dependency across other quantities and levels), the use of copula-based synthetic data generation may prove useful to make emulators more resistant to outliers (e.g. in climate change scenario settings) and should be investigated in future research.



**Code and data availability**

The key software used in this paper are Synthia (available under MIT licence at https://github.com/dmey/synthia) and TensorFlow (available under Apache 2.0 licence at https://www.tensorflow.org/). Software, data, and tools are archived with a Singularity (Kurtzer et al., 2017) image deposited on Zenodo as described in the scientific reproducibility section of Meyer

et al. (2020). Users wishing to download or reproduce the results described in this paper can download the archive at https://doi.org/10.5281/zenodo.4320795 and optionally run Singularity on their local or remote systems.

**Author contribution**

Conceptualization, D.M.; Data curation, D.M.; Formal analysis, D.M., T.N.; Investigation, D.M.; Methodology, D.M., T.N., R.H.; Software, D.M.; Resources, D.M.; Validation, D.M.; Visualization, D.M.; Writing – original draft preparation, D.M., T.N.;

Writing – review & editing, D.M., T.N., R.H..

**Competing interests**

The authors declare no conflict of interest.

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
