# Peer review of "Copula-based synthetic data generation for machine learning emulators in weather and climate: application to a simple radiation model"

_Geoscientific Model Development, 2020_

## Referee Comment (RC2)

Specific Reviewer comments for "Copula-based synthetic data generation for machine learning emulators in weather and climate: application to a simple radiation model"

**Major comments**

Section 2.1 defines a general methodology for training ML models using either Observation-based training (OBT) or Emulation-based training (EBT). However, it appears that these definitions are inconsistent with the methods used in the paper. From what I can work out, the different methods are:

A.  OBT (as defined): Train ML model on inputs X and outputs Y. *The source of X and Y is not defined, but is presumably observations, pseudo-observations or the like.*
A.  OBT (as implemented): ***Inconsistent with A (as defined), since Y comes from a physical model fed with X like in method C (EBT). In other words, A is not implemented, only C.***

B.  OBT with data generation (as defined): Train ML model on inputs X' and outputs Y', where X' and Y' is synthetic data created using a data generation model (copula) fitted to X, Y. *The source of X and Y is not defined, but is presumably observations, pseudo-observations or the like.*
B.  OBT with data generation (as implemented): ***Inconsistent with B (as defined), since Y comes from a physical model fed with X like in EBT.***

C.  EBT (as defined): Train ML model on inputs X and outputs Y, where Y has been generated by feeding inputs X to a physical model.
C.  EBT (as implemented): ***Consistent with C (as defined), X comes from satellite data and Y comes from a toy physical model fed with X.***

D.  EBT with data generation (as defined): Train ML model on inputs X' and outputs Y', where inputs X' have been created using a data generation model (copula) fitted to X, and outputs Y' come from a physical model fed with X'.
D.  EBT with data generation (as implemented):  **Otherwise consistent with D (as defined), but X and Y are included in training data.**

Claims to study the use of synthetic data for OBT in this paper are weak in the absence of observationally sourced outputs. If such data is unavailable or obtaining it is outside the scope of this work then my suggestion then the methodology should be reframed. Alternatively, make it clear that B is implemented in such a way to only mimic the defined method, in the absence of suitable data, and argue why this is valid.

Figure 1 is also inconsistent with the methods used. This figure should either be changed to describe the actual implementation, or a second figure with those could be made, something like this:

| Baseline | Synthetic data method 1 (D) | Synthetic data method 2 (B) |
| --- | --- | --- |
| X | X | X |
| \|
 \|
 \| | X
 generative model
 X' | X
 physical model
 Y |
| X
 physical model
 Y | X+X'
 physical model
 Y+Y' | X, Y
 generative model
 X', Y' |
| X, Y
 prediction model | (X+X'),(Y+Y')
 prediction model using
 synthetic inputs | (X+X'),(Y+Y')
 prediction model using
 synthetic inputs and outputs |

**Minor comments:**

**L25:** Chevallier and Krasnopolsky laid out the initial work but newer studies should be mentioned, e.g. Ukkonen et al. (2020) and Veerman et al. (2021). A lot of work has also been done recently to train NNs on data from cloud-resolving models (e.g. Brenowitz et al. 2018, Gentine et al. 2018, Rasp et al. 2018), which I'm not sure if it fits the authors definition of EBT (since X and Y has the same source) but it's important to place the present work in the context of the wider literature.

**L35-46; section 2.1**: another approach for "EBT with data generation", which is probably worth mentioning, is to simply sample the multidimensional input space rigorously using either random methods such as Latin hypercube sampling, or deterministic methods such as a Halton sequence. As demonstrated by Ukkonen et al. (2020), who sampled gas concentrations uniformly while keeping the dependencies of pressure and temperature intact, this can work well for difficult problems where generating outputs with physical models is cheap (EBT).

Whether it's better to train ML models on "realistic" datasets where the observed dependencies between inputs are respected, or on "dense and wide" data which sample the input space (N-dimensional hypercube) more uniformly, is to me nontrivial. On one hand, the latter may result in more generalizable models which also learn the underlying physics more effectively, assuming that the input-output mapping given by the physical model remains to some extent rooted in physics throughout this expanded input space. On the other hand, it may come at a significant cost in model complexity and computational resources. It's even possible that minimizing errors across the domain space comes at the expense of degraded performance on real datasets, regardless of model complexity, due to imperfect information. This would certainly encourage the use of approaches presented here, e.g. copulas, which respect the observed statistical distributions. (If the authors are aware of any studies on this I would be interested to know.)

**L69:** Before seeing the results, is strategy B a legitimate approach? It may be useful but I have not seen it being used. Fitting a simpler statistical model (copula) on observations and using it to create synthetic inputs and outputs to train a more complex statistical model (ML) seems a bit odd - does it actually extract any new information?

**L85:** The use of cloud optical depth instead of total optical depth for predicting longwave radiation only makes sense when the outputs come from a toy model and not observations, since clear-sky absorption is important for observed long-wave radiation. Again, due to the confusing overview of methods (2.1) the unobservant reader might think a method where the outputs come from observations is included, in which case the chosen inputs appear strange.

**L90**: *"We then define case A (or C) as the baseline.."* Here it first becomes apparent that both A and C are **not** used in this paper. The authors seem to redefine A to be equal to C, as they do not have observations to use as target outputs for Observation-based training (A), but then it should be made clear that strategy A is in fact not implemented.

**Sections 2.3.2 - 2.3.4**: For someone who was unfamiliar with copulas, this served as a good and clear introduction for the most part, but I am left confused about what kind of assumptions/parameters the "Vine-parametric" copula uses to model the dependence of two variables?

**L230:** *"using the source X or augmented X' data depending on the strategy (i.e. OBT or EBT)."* Again, confusing - OBT is not actually implemented. Furthermore, in Figure 1, X and X' are used in both strategy B (OBT-Augmented) and strategy D (EBT-Augmented), which is inconsistent with the highlighted sentence.

**Figure 3.** Perhaps a diagonal 1:1 line would aid interpretation, but this is a matter of style.

**Figure 5 d).** These results are good and quite interesting.

References:
Brenowitz, N. D., & Bretherton, C. S. (2018). Prognostic validation of a neural network unified physics parameterization. *Geophysical Research Letters*, *45*(12), 6289-6298.
Gentine, P., Pritchard, M., Rasp, S., Reinaudi, G., & Yacalis, G. (2018). Could machine learning break the convection parameterization deadlock?. *Geophysical Research Letters*, *45*(11), 5742-5751.
Pal, A., Mahajan, S., & Norman, M. R. (2019). Using deep neural networks as cost-effective surrogate models for super-parameterized E3SM radiative transfer. *Geophysical Research Letters*, *46*(11), 6069-6079.
Rasp, S., Pritchard, M. S., & Gentine, P. (2018). Deep learning to represent subgrid processes in climate models. *Proceedings of the National Academy of Sciences*, *115*(39), 9684-9689.
Ukkonen, P., Pincus, R., Hogan, R. J., Pagh Nielsen, K., & Kaas, E. (2020). Accelerating radiation computations for dynamical models with targeted machine learning and code optimization. *Journal of Advances in Modeling Earth Systems*, *12*(12), e2020MS002226.
Veerman, M. A., Pincus, R., Stoffer, R., van Leeuwen, C. M., Podareanu, D., & van Heerwaarden, C. C. (2021). Predicting atmospheric optical properties for radiative transfer computations using neural networks. *Philosophical Transactions of the Royal Society A*, *379*(2194), 20200095.

---

## Author Comment (AC2)

We thank reviewers for their useful comments and appreciation of our work. After careful consideration of all comments and feedback provided, we opted to rewrite large sections of the introduction and methods to best improve the overall clarity. In the following we respond to each point individually. Wherever necessary, our responses (RX.X) give the updated manuscript's new line numbers as LXX. To remove duplication, reviewers' reference lists have been merged into a single global reference list at the end of this document and, where necessary, their in-text citations have been edited in place. No other changes to reviewers' original comments have been made.

**Reviewer #1**

The title of the paper is: "Copula-based synthetic data generation for machine learning emulators in weather and climate: application to a simple radiation model". The major conclusion of the paper is that using emulated data may be of particular interest to scientists and practitioners developing ML emulators in weather and climate. In my opinion both the title and the main conclusion are unfocused and the main conclusion is not something new.

First of all, authors should specify how they understand "machine learning emulators in weather and climate". There is a long history of developing ML emulators for numerical weather and climate models, including emulations of sophisticated state-of-the-art radiation models (Chevallier et al., 2000; Krasnopolsky et al., 2005, 2008, 2010; Brenowitz and Bretherton, 2018; Gentine et al., 2018; O'Gorman and Dwyer, 2018; Pal et al., 2019; Rasp et al., 2018; Scher and Messori, 2019; Scher, 2018; Rasp, 2020) plus a quickly growing amount of publications during 2019, 2020, and even already during 2021. In all these works, data simulated by GCM, CRM, or LES are used. Researchers, working in these fields of weather and climate, long ago recognized that they must use simulated data because there is no sufficient amount of observations, collocated in space and time to use for training their ML emulations. In some cases ECMWF or NCEP analysis or reanalysis are used to integrate in an optimal way simulated and observed data. The same sources of data are used to develop forward models.

Thus, I suggest to authors:
1. to clearly specify what subfield of "weather and climate" they target
**R1.1:** We believe that the lack of clarity in the introduction and method section may have led to some misunderstandings. The main misunderstanding seems to be caused by the term "data generation" and "emulator/emulation". By emulator/emulation we mean a machine learning (ML) model which is a surrogate of physical one (e.g. Chevallier et al., 1998; Meyer et al., 2021; Ukkonen et al., 2020). The focus of this paper is on the cheap generation of more inputs for the physical model as a way to generate many more input-output pairs for training a ML emulator, thereby improving the resulting emulator. When we say cheap we mean much cheaper than, for example, running a full GCM (global climate model), CRM (cloud resolving model) or LES (large eddy simulation) to produce inputs for the physical model such as a radiation scheme. Naturally, using "simulated" observations produced by a full atmospheric model such as a GCM is a very common approach in producing training datasets for ML, and would not form the basis for an original paper. The novelty and originally of our work lies in being able to generate new input samples for physical models which are statistically similar to the original samples and thus representative of the original data (e.g. absolute temperature within ranges found in the Earth's atmosphere, and similar temperature values between two adjacent grid points). After this the physical model is run with the original and augmented input dataset to generate output pairs which are eventually used to train the ML emulator. In our paper to show whether data generation has a positive or a negative impact we show results from predictions made with ML emulators trained with augmented datasets against those trained with the original dataset.

As mentioned earlier, although there may have been misunderstandings with this paper, we fully appreciate that reviewer's comments and have therefore narrowed the focus to just consider the case of ML emulators and rewritten large sections of the introduction, methods and abstract to convey more clarity.

With regards to the theoretical basis to limit the use of data generation to any particular subfield of weather and climate, this method is best suited when the number of samples is much larger than the number of features as models scale quadratically in the number of features – in terms of both memory and computational complexity (please see L327-333 for more details).

2. to edit the title and the text correspondingly
**R1.2:** This appear to be related with our reply in R1.1 that there may have been a misinterpretation of the term "data generation" and emulators which we hope to have clarified in R1.1. We appreciate that the previous title may have been too verbose/unfocused. To improve clarity we have therefore edited the title to "Copula-based synthetic data augmentation for machine learning emulators" which we think is a better representation of our work. Here we use the word "augmentation" to indicate statistical generation of data, to contrast with "simulation" which could encompass running a full GCM to generate data (at much more expense). With regards to the text we have heavily reworded the abstract and introduction/method sections as mentioned in R1.1.

3. to explain how their results provide researchers, working in the targeted field, with a new information

**R1.3:** This remark is likely caused by a possible misinterpretation of our work. As outlined in R1.1 and R1.2 we appreciate the reviewer's feedback and have therefore heavily reworded the abstract introduction/method sections to improve clarity and focus. We believe that the updated manuscript demonstrates our work to be novel. We are unaware of similar work but would welcome pointers to similar work in the literature. We believe that the results and conclusion as outlined in the updated version provide readers with new ways to improve emulators in new ways.

**Reviewer #2**

The authors have written an interesting paper on the use of copulas for synthetic data generation for ML emulators in weather and climate applications. This is a valuable and, as far as I know, novel contribution to the ML in weather and climate modelling field. The paper is well-written and for the most part, clear and concise.

Besides some improvements that could be made with regards to literature review, I found a larger issue with regards to the methodology, which appears inconsistent in the way it is defined and implemented. The authors initially appear to use both observation-based training (OBT) and emulation-based training (EBT), but strictly speaking no OBT is done as the outputs come from a physical model (just like in the authors' definition of EBT). Perhaps the implementation of method B (OBT with data generation) is intended only to mimic the method as it is defined, but then this should be made explicit and the validity of this approach discussed.

While some readers may find no problem with the conceptualization and presentation, since it is open to interpretation, I found it nonintuitive. That being said, this a publishable paper if the clarity is improved. I attach my specific comments as a PDF supplement.

**Major comments**
Section 2.1 defines a general methodology for training ML models using either Observation-based training (OBT) or Emulation-based training (EBT). However, it appears that these definitions are inconsistent with the methods used in the paper. From what I can work out, the different methods are:

A. OBT (as defined): Train ML model on inputs X and outputs Y. The source of X and Y is not defined, but is presumably observations, pseudo-observations or the like.
A. OBT (as implemented): **Inconsistent with A (as defined), since Y comes from a physical model fed with X like in method C (EBT). In other words, A is not implemented, only C.**

B. OBT with data generation (as defined): Train ML model on inputs X' and outputs Y', where X' and Y' is synthetic data created using a data generation model (copula) fitted to X, Y. *The source of X and Y is not defined, but is presumably observations, pseudo-observations or the like.*
B. OBT with data generation (as implemented): *Inconsistent with B (as defined), since Y comes from a physical model fed with X like in EBT.*

C. EBT (as defined): Train ML model on inputs X and outputs Y, where Y has been generated by feeding inputs X to a physical model.
C. EBT (as implemented): **Consistent with C (as defined), X comes from satellite data and Y comes from a toy physical model fed with X.**

D. EBT with data generation (as defined): Train ML model on inputs X' and outputs Y', where inputs X' have been created using a data generation model (copula) fitted to X, and outputs Y' come from a physical model fed with X'.
D. EBT with data generation (as implemented): **Otherwise consistent with D (as defined), but X and Y are included in training data.**

Claims to study the use of synthetic data for OBT in this paper are weak in the absence of observationally sourced outputs. If such data is unavailable or obtaining it is outside the scope of this work then my suggestion then the methodology should be reframed. Alternatively, make it clear that B is implemented in such a way to only mimic the defined method, in the absence of suitable data, and argue why this is valid.

Figure 1 is also inconsistent with the methods used. This figure should either be changed to describe the actual implementation, or a second figure with those could be made, something like this:

| Baseline | Synthetic data method 1 (D) | Synthetic data method 2 (B) |
|----------|------------------------------|------------------------------|
| X | X | X |

```
* * *
      |                          X                          X
      |                   generative model            physical model
      |                         X'                          Y
* * *
      X                        X+X'                        X, Y
 physical model           physical model            generative model
      Y                        Y+Y'                        X', Y'
* * *
     X, Y                   (X+X'),(Y+Y')             (X+X'),(Y+Y')
 prediction model        prediction model using     prediction model using
                         synthetic inputs           synthetic inputs and outputs
* * *
```

**R2.1:** Together with R1.1 we have simplified the methodology to now only include 2 strategies from the 4 mentioned previously and have amended Figure 1 as suggested (please see the revised sections 1-2). The training now consists of a clear baseline defined as inputs X fed to the physical model to generate corresponding outputs Y; then X and Y are used to train the ML emulator (please see Figure 1 A). The augmentation strategy is now only one (i.e. we have removed "strategy B" mentioned in the previous version of the manuscript; "synthetic data method 2" in Figure 1 above) The only augmentation strategy uses three different copula models to fit inputs X to generate synthetic inputs X'; then inputs X and X' are used as inputs to the physical model to generate the corresponding outputs Y and Y'; finally both X and X', and Y and Y' are used to train the ML emulator (please see Figure 1 B).

**Minor comments**
**L25:** Chevallier and Krasnopolsky laid out the initial work but newer studies should be mentioned, e.g. Ukkonen et al. (2020) and Veerman et al. (2021). A lot of work has also been done recently to train NNs on data from cloud-resolving models (e.g. Brenowitz and Bretherton 2018; Gentine et al. 2018; Rasp et al. 2018), which I'm not sure if it fits the authors definition of EBT (since X and Y has the same source) but it's important to place the present work in the context of the wider literature.
**R2.2:** We have added more literature to the introduction as suggested, please see L18-24.

**L35-46; section 2.1**: another approach for "EBT with data generation", which is probably worth mentioning, is to simply sample the multidimensional input space rigorously using either random methods such as Latin hypercube sampling, or deterministic methods such as a Halton sequence. As demonstrated by Ukkonen et al. (2020), who sampled gas concentrations uniformly while keeping the dependencies of pressure and temperature intact, this can work well for difficult problems where generating outputs with physical models is cheap (EBT). Whether it's better to train ML models on "realistic" datasets where the observed dependencies between inputs are respected, or on "dense and wide" data which sample the input space (Ndimensional hypercube) more uniformly, is to me nontrivial. On one hand, the latter may result in more generalizable models which also learn the underlying physics more effectively, assuming that the input output mapping given by the physical model remains to some extent rooted in physics throughout this expanded input space. On the other hand, it may come at a significant cost in model complexity and computational resources. It's even possible that minimizing errors across the domain space comes at the expense of degraded performance on real datasets, regardless of model complexity, due to imperfect information. This would certainly encourage the use of approaches presented here, e.g. copulas, which respect the observed statistical distributions. (If the authors are aware of any studies on this I would be interested to know.)
**R2.3:** The use of Latin hypercube sampling (LHS) strictly refers to sampling. LHS could be performed after generating data using copulas. With regards to the best approach to use, it generally depends on the application. For example, sometimes it may be preferable to improve the skills over a wide range of parameters at the cost of others by modifying or "uniformifying" specific marginal distribution. We allow for these cases to be easily modelled through the Synthia tool. Similarly to Ukkonen et al. (2020) in Meyer et al. (2021) we also augment the number of vertical profiles by generating independent samples from original distributions of albedo and solar zenith angle using Synthia.

**L69:** Before seeing the results, is strategy B a legitimate approach? It may be useful but I have not seen it being used. Fitting a simpler statistical model (copula) on observations and using it to create synthetic inputs and outputs to train a more complex statistical model (ML) seems a bit odd - does it actually extract any new information?
**R2.4:** Because of changes made in R1.1 and as mentioned in R2.1, "strategy B" mentioned in the previous version of the manuscript (or "synthetic data method 2" in Figure 1 in R2.1) has been removed. However, to answer the reviewer's question this method is generally used to address confidentiality or licensing restrictions (e.g. Patki et al., 2016). Although this was not the focus of the paper, results in the original manuscript showed that when applied to this type of data the approach my lead to a large deterioration in accuracy.

**L85:** The use of cloud optical depth instead of total optical depth for predicting longwave radiation only makes sense when the outputs come from a toy model and not observations, since clear-sky absorption is important for observed long-wave radiation.

Again, due to the confusing overview of methods (2.1) the unobservant reader might think a method where the outputs come from observations is included, in which case the chosen inputs appear strange.

**R2.5:** As explained just before Eq. 2, clear-sky absorption is treated approximately in the toy model as a column optical depth of 1.7, which is the best fit to the full radiation scheme. As this is assumed not to vary from profile to profile, it is not listed as an "input" in section 2.1, but we now point this out in section 2.1 (L78-79).

**L90**: "*We then define case A (or C) as the baseline..*" Here it first becomes apparent that both A and C are **not** used in this paper. The authors seem to redefine A to be equal to C, as they do not have observations to use as target outputs for Observation-based training (A), but then it should be made clear that strategy A is in fact not implemented.

**R2.6:** Because of changes made in R1.1 this has now been removed and no longer applies.

**Sections 2.3.2 - 2.3.4**: For someone who was unfamiliar with copulas, this served as a good and clear introduction for the most part, but I am left confused about what kind of assumptions/parameters the "Vine-parametric" copula uses to model the dependence of two variables?

**R2.7:** We have added more information about fitting in section 2.3.4 please see L179-182.

**L230**: "*using the source X or augmented X' data depending on the strategy (i.e. OBT or EBT).*" Again, confusing - OBT is not actually implemented. Furthermore, in Figure 1, X and X' are used in both strategy B (OBT-Augmented) and strategy D (EBT-Augmented), which is inconsistent with the highlighted sentence.

**R2.8:** Because of changes made in R1.1 this has now been removed and no longer applies.

**Figure 3.** Perhaps a diagonal 1:1 line would aid interpretation, but this is a matter of style.

**R2.9:** We have amended Figure 3 accordingly.

**Figure 5 d).** These results are good and quite interesting.

**R2.10**: Thank you.

**Reviewer #3**

**Comments:**
- It is my impression that the nomenclature Emulator-based training does not accurately reflect the principles of what is done in the setting described by the authors. Independently of how we train our ML models (using observations or simulations), both general settings provide a surrogate (an emulator) to make predictions. I believe the term "Simulation-based training (SBT)" reflects better the principles of training a surrogate model based on simulations being done by, what the authors call, physical models. This distinction, although a bit superficial, has been carefully proposed in earlier work in general scientific endeavors. See for example: Jerome Sacks et al. (1989), which has led to a more mature framework of modeling emulators based on simulation results to predict real world processes. See: Kennedy and O'Hagan (2001).

**R3.1:** We appreciate that this may have been confusing. Together with R1.1 and R2.1 we made extensive changes in the method sections to improve clarity. We now only refer to a baseline (corresponding to Figure 1 C in the previous version of the manuscript) and an augmentation strategy (corresponding to Figure 1 D in the previous version of the manuscript). We have removed references to observation- and emulation-/simulation- based strategies as are no longer used. Please see R1.1 and R2.1 and revised sections 1 and 2 for more details.

- I believe that to refer to the parameters of a ML model as _weights_ lessens the generalizability of the work of the authors. This is because the term "weight" is particular to Deep Learning (DL) instances such as Artificial Neural Networks or Multilayer Linear Perceptron models and their subsequent generalizations. My issue with this is that ML is a much broader discipline than just DL. *Any* ML used for prediction is looking for the best possible association of $X$ (features, properties, descriptors) to $Y$ (the target). In this sense, we are looking for the best candidate $h$ that can achieve $Y \approx h(X)$ in some sense (for example, as measured by Mean-Square Error). We choose parametrized models due to our ability ---mainly, through iterative optimization algorithms--- to learn such approximations. Although, for an ML researcher/practitioner this is not new. Someone with no such background would not make the immediate connection in the text with $w$ to "the best function approximation for a specific model architecture".

**R3.2:** We agree. We have rephrased this (L69-70) and removed references to the term "weight" and referred to outputs as "m" for model in Figure 1.

- This leads me to my next concern: are simulations contrasted to data? Even though, a surrogate is built upon simulation-based observations, at some point it needs to be contrasted to real world data to measure the validity of using a specific model configuration (either for the physical model or the emulator). There is work being done in this direction for climate predictions. See, for example: Cleary et al. (2021) as an example of a complete simulation-observation based strategy to learn surrogate models. The goal of learning ML models for climate applications is a hot topic in research. You can also see: Rasp et al. (2018)

**R3.3:** The question of how emulators perform against real data is a separate question to that of whether synthetic data improve predictions of ML emulators. Previous work on radiation emulation (e.g. Chevallier et al., 1998; Krasnopolsky et al., 2010; Ukkonen et al., 2020; Veerman et al., 2021) also compared their results against pseudo-observations. Furthermore, as in radiation one cannot easily measure atmospheric fluxes precisely (i.e. on average closer than 5 W m$^{-2}$), and as done in similar studies we are only using a very simple toy to showcase the general method (e.g. Dueben and Bauer, 2018; Rasp, 2020), we believe that evaluating our results against real data is out of scope. The scope of this paper is to show whether emulators trained with synthetically augmented dataset have lower prediction errors than a baseline emulator trained on the original dataset.

Which leads me to question what is the novelty of the proposed manuscript.
**R3.4:** We believe this to be a misunderstanding due to a lack of clarity, please see R1.1 and R1.3.

- The stochasticity of the training of an MLP can be accounted, for example, with $k$-fold cross validation. It should also be used as it internally provides a measure of generalization error that can help guide the selection of certain hyper-parameters (for example the optimization-related parameters). Why not use such a strategy for this manuscript?
**R3.5:** Cross validation is generally useful for small datasets; in our case the size of samples is reasonably large. The smallest case (i.e. without data generation) consists of 25 000 profiles x 137 levels = 3 425 000 samples or 1 370 000/685 000/1 370 000 with train/validation/test split. Repeating training 10 times to account for the stochasticity at training is a simple and effective method that allow for meaningful statistics to be computed and therefore we do not see the need to use cross validation in this instance.

- As a final comment, it is not clear to me the setting and the intention of the strategy presented by the authors. I believe it is not clear how to interpret these results. It seems like a particular instance of a data-augmentation strategy, with the potential benefit of preserving the observed probabilistic relationships among training data. I believe the authors does not provide clear evidence that such an augmentation achieves better results when compared to real world observations.
**R3.6:** We believe this to be a misunderstanding due to a lack of clarity, please see R1.1.

**Minor comments:**
- Typo found at line 151. It reads "...all variables a continuous..." it should be "...all variables are continuous...".
**R3.7:** Fixed.

**References**

Brenowitz, N. D. and Bretherton, C. S.: Prognostic Validation of a Neural Network Unified Physics Parameterization, Geophys. Res. Lett., 45, 6289–6298, https://doi.org/10.1029/2018GL078510, 2018.

Chevallier, F., Ruy, F. C., Scott, N. A., and Din, A. C.: A Neural Network Approach for a Fast and Accurate Computation of a Longwave Radiative Budget, 37, 13, https://doi.org/10.1175/1520-0450(1998)037<1385:ANNAFA>2.0.CO;2, 1998.

Chevallier, F., Morcrette, J.-J., Chéruy, F., and Scott, N. A.: Use of a neural-network-based long-wave radiative-transfer scheme in the ECMWF atmospheric model, Q.J.R. Meteorol. Soc., 126, 761–776, https://doi.org/10.1002/qj.49712656318, 2000.

Cleary, E., Garbuno-Inigo, A., Lan, S., Schneider, T., and Stuart, A. M.: Calibrate, emulate, sample, Journal of Computational Physics, 424, 109716, https://doi.org/10.1016/j.jcp.2020.109716, 2021.

Dueben, P. D. and Bauer, P.: Challenges and design choices for global weather and climate models based on machine learning, Geosci. Model Dev., 11, 3999–4009, https://doi.org/10.5194/gmd-11-3999-2018, 2018.

Gentine, P., Pritchard, M., Rasp, S., Reinaudi, G., and Yacalis, G.: Could Machine Learning Break the Convection Parameterization Deadlock?, Geophys. Res. Lett., 45, 5742–5751, https://doi.org/10.1029/2018GL078202, 2018.

Jerome Sacks, William J. Welch, Toby J. Mitchell, and Henry P. Wynn: Design and Analysis of Computer Experiments, Statistical Science, 4, 409–423, https://doi.org/10.1214/ss/1177012413, 1989.

Kennedy, M. C. and O'Hagan, A.: Bayesian calibration of computer models, J Royal Statistical Soc B, 63, 425–464, https://doi.org/10.1111/1467-9868.00294, 2001.

Krasnopolsky, V. M., Fox-Rabinovitz, M. S., and Chalikov, D. V.: New Approach to Calculation of Atmospheric Model Physics: Accurate and Fast Neural Network Emulation of Longwave Radiation in a Climate Model, Mon. Wea. Rev., 133, 1370–1383, https://doi.org/10.1175/MWR2923.1, 2005.

Krasnopolsky, V. M., Fox-Rabinovitz, M. S., and Belochitski, A. A.: Decadal Climate Simulations Using Accurate and Fast Neural Network Emulation of Full, Longwave and Shortwave, Radiation, Mon. Wea. Rev., 136, 3683–3695, https://doi.org/10.1175/2008MWR2385.1, 2008.

Krasnopolsky, V. M., Fox-Rabinovitz, M. S., Hou, Y. T., Lord, S. J., and Belochitski, A. A.: Accurate and Fast Neural Network Emulations of Model Radiation for the NCEP Coupled Climate Forecast System: Climate Simulations and Seasonal Predictions, Mon. Wea. Rev., 138, 1822–1842, https://doi.org/10.1175/2009MWR3149.1, 2010.

Meyer, D., Hogan, R. J., Dueben, P. D., and Mason, S. L.: Machine Learning Emulation of 3D Cloud Radiative Effects, https://arxiv.org/abs/2103.11919, 2021.

O'Gorman, P. A. and Dwyer, J. G.: Using Machine Learning to Parameterize Moist Convection: Potential for Modeling of Climate, Climate Change, and Extreme Events, J. Adv. Model. Earth Syst., 10, 2548–2563, https://doi.org/10.1029/2018MS001351, 2018.

Pal, A., Mahajan, S., and Norman, M. R.: Using Deep Neural Networks as Cost-Effective Surrogate Models for Super-Parameterized E3SM Radiative Transfer, Geophys. Res. Lett., 46, 6069–6079, https://doi.org/10.1029/2018GL081646, 2019.

Patki, N., Wedge, R., and Veeramachaneni, K.: The Synthetic Data Vault, in: 2016 IEEE International Conference on Data Science and Advanced Analytics (DSAA), 2016 IEEE International Conference on Data Science and Advanced Analytics (DSAA), Montreal, QC, Canada, 399–410, https://doi.org/10.1109/DSAA.2016.49, 2016.

Rasp, S.: Coupled online learning as a way to tackle instabilities and biases in neural network parameterizations: general algorithms and Lorenz 96 case study (v1.0), Geosci. Model Dev., 13, 2185–2196, https://doi.org/10.5194/gmd-13-2185-2020, 2020.

Rasp, S., Pritchard, M. S., and Gentine, P.: Deep learning to represent subgrid processes in climate models, Proc Natl Acad Sci USA, 115, 9684–9689, https://doi.org/10.1073/pnas.1810286115, 2018.

Scher, S.: Toward Data-Driven Weather and Climate Forecasting: Approximating a Simple General Circulation Model With Deep Learning, Geophys. Res. Lett., 45, 12,616-12,622, https://doi.org/10.1029/2018GL080704, 2018.

Scher, S. and Messori, G.: Weather and climate forecasting with neural networks: using GCMs with different complexity as study-ground, Atmospheric Sciences, https://doi.org/10.5194/gmd-2019-53, 2019.

Ukkonen, P., Pincus, R., Hogan, R. J., Nielsen, K. P., and Kaas, E.: Accelerating radiation computations for dynamical models with targeted machine learning and code optimization, 12, https://doi.org/10.1029/2020ms002226, 2020.

Veerman, M. A., Pincus, R., Stoffer, R., van Leeuwen, C. M., Podareanu, D., and van Heerwaarden, C. C.: Predicting atmospheric optical properties for radiative transfer computations using neural networks, Phil. Trans. R. Soc. A., 379, 20200095, https://doi.org/10.1098/rsta.2020.0095, 2021.